# The histone H3K9 demethylase KDM3A promotes anoikis by transcriptionally activating pro-apoptotic genes *BNIP3* and *BNIP3L*

Victoria E Pedanou[1,2], Stéphane Gobeil[3,4], Sébastien Tabariès[5], Tessa M Simone[1,2], Lihua Julie Zhu[1,6,7], Peter M Siegel[5], Michael R Green[1,2]*

[1]Department of Molecular, Cell and Cancer Biology, University of Massachusetts Medical School, Worcester, United States; [2]Howard Hughes Medical Institute, University of Massachusetts Medical School, Worcester, United States; [3]Department of Molecular Medicine, Université Laval, Quebec City, Canada; [4]Centre de recherche du CHU de Québec, CHUL, Québec PQ, Canada; [5]Department of Medicine, Goodman Cancer Research Centre, McGill University, Montreal, Canada; [6]Program in Molecular Medicine, University of Massachusetts Medical School, Worcester, United States; [7]Program in Bioinformatics and Integrative Biology, University of Massachusetts Medical School, Worcester, United States

*For correspondence: michael. green@umassmed.edu

**Competing interests:** The authors declare that no competing interests exist.

**Abstract** Epithelial cells that lose attachment to the extracellular matrix undergo a specialized form of apoptosis called anoikis. Here, using large-scale RNA interference (RNAi) screening, we find that KDM3A, a histone H3 lysine 9 (H3K9) mono- and di-demethylase, plays a pivotal role in anoikis induction. In attached breast epithelial cells, *KDM3A* expression is maintained at low levels by integrin signaling. Following detachment, integrin signaling is decreased resulting in increased *KDM3A* expression. RNAi-mediated knockdown of *KDM3A* substantially reduces apoptosis following detachment and, conversely, ectopic expression of *KDM3A* induces cell death in attached cells. We find that KDM3A promotes anoikis through transcriptional activation of *BNIP3* and *BNIP3L*, which encode pro-apoptotic proteins. Using mouse models of breast cancer metastasis we show that knockdown of *Kdm3a* enhances metastatic potential. Finally, we find defective *KDM3A* expression in human breast cancer cell lines and tumors. Collectively, our results reveal a novel transcriptional regulatory program that mediates anoikis.

## Introduction

Epithelial cells that lose attachment to the extracellular matrix (ECM), or attach to an inappropriate ECM, undergo a specialized form of apoptosis called anoikis. Anoikis has an important role in preventing oncogenesis, particularly metastasis, by eliminating cells that lack proper ECM cues (*Simpson et al., 2008*; *Zhu et al., 2001*). Anoikis also functions to prevent the invasion of tumor cells into the luminal space, which is a hallmark of epithelial tumors (*Debnath et al., 2002*). In general, epithelial-derived cancers, such as breast cancer, develop resistance to anoikis (reviewed in *Schwartz, 1997*). Several signaling pathways have been shown to regulate anoikis (reviewed in *Paoli et al., 2013*). In particular, anoikis is suppressed by integrin signaling, which functions through focal adhesion kinase (FAK), an activator of the RAF/MEK/ERK pathway (*King et al., 1997*). FAK signaling is active in attached cells and is inactive following detachment (*Frisch et al., 1996*). Anoikis is also suppressed by integrin-mediated, ligand independent activation of the epidermal growth factor

**eLife digest** Epithelial cells line the inside of blood vessels, intestines and other organs throughout the body. Any epithelial cells that become detached from their natural surroundings die by a process called anoikis (a Greek word meaning "being without a home"). This process has an important role in preventing cancer from spreading around the body because it eliminates cells that are not in their proper environment. However, some cancers that start from epithelial cells, such as breast cancer, develop resistance to anoikis. Gaining a better understanding of the cellular factors that regulate anoikis, and how resistance develops, may reveal new drug targets for the treatment of breast cancer.

Previous studies found proteins called BIM and BMF promote anoikis by inducing cell suicide. However, it is possible that other factors can also promote this process in different ways. Pedanou et al. performed a large-scale genetic screen in human breast epithelial cells and identified several new factors that promote anoikis.

Inside our cells, DNA is packaged around proteins called histones, which can influence whether a gene is switched on or off. One of the factors Pedanou et al. identified is a protein called KDM3A that can remove small chemical groups (known as methyl groups) from histones – a process that is known to switch on genes. Further experiments show that epithelial cells in their natural surroundings only produce low levels of KDM3A, but that the levels of this protein increase if these cells become detached. This promotes anoikis by activating two genes called *BNIP3* and *BNIP3L* that induce cell suicide. However, KDM3A levels are low in human breast cancers, which suggests that these cancers become resistant to anoikis by preventing increases in KDM3A production.

Using a mouse model of breast cancer, Pedanou et al. found that switching off KDM3A in cancer cells increases their ability to move around the body. Collectively, these findings reveal a new mechanism that triggers anoikis in normal breast epithelial cells and is disabled during breast cancer development. Future challenges are to identify factors that directly regulate the production of KDM3A, and to understand how these factors are manipulated in breast cancer cells to cause anoikis resistance.

receptor (EGFR) signaling pathway (*Moro et al., 1998*), which, like FAK, also stimulates RAF/MEK/ERK activity.

These cell signaling pathways have been found to regulate the levels of BIM (also called BCL2L11) and BMF, two pro-apoptotic members of the BCL2 family of apoptosis regulators previously shown to contribute to anoikis (*Reginato et al., 2003*; *Schmelzle et al., 2007*). However, depletion of BIM or BMF diminishes but does not completely prevent anoikis (*Reginato et al., 2003*; *Schmelzle et al., 2007*), suggesting the existence of other factors and regulatory pathways that can promote anoikis. Moreover, the basis of anoikis resistance remains to be determined and to date has not been linked to alterations in expression or activity of BIM or BMF.

## Results and discussion

To investigate the possibility that there are additional factors and regulatory pathways that promote anoikis, we performed a large-scale RNA interference (RNAi) screen for genes whose loss of expression confer anoikis resistance. The screen was performed in MCF10A cells, an immortalized but non-transformed human breast epithelial cell line that has been frequently used to study anoikis (see, for example, *Huang et al., 2010*; *Reginato et al., 2003*; *Schmelzle et al., 2007*; *Taube et al., 2006*). A genome-wide human small hairpin RNA (shRNA) library comprising ~62,400 shRNAs directed against ~28,000 genes (*Silva et al., 2003*; *Silva et al., 2005*) was divided into 10 pools, which were packaged into retroviral particles and used to stably transduce MCF10A cells. Following selection, the cells were divided into two populations, one of which was plated on poly-2-hydroxyethylmethacry-late (HEMA)-coated plates for 10 days to inhibit cell attachment to matrix, and another that was cultured attached to matrix for 10 days as a control (*Figure 1A*). Surviving cells were selected and shRNAs identified by deep sequencing. Bioinformatic analysis of the two populations identified 26 shRNAs whose abundance was significantly enriched >500-fold following detachment (*Figure 1—*

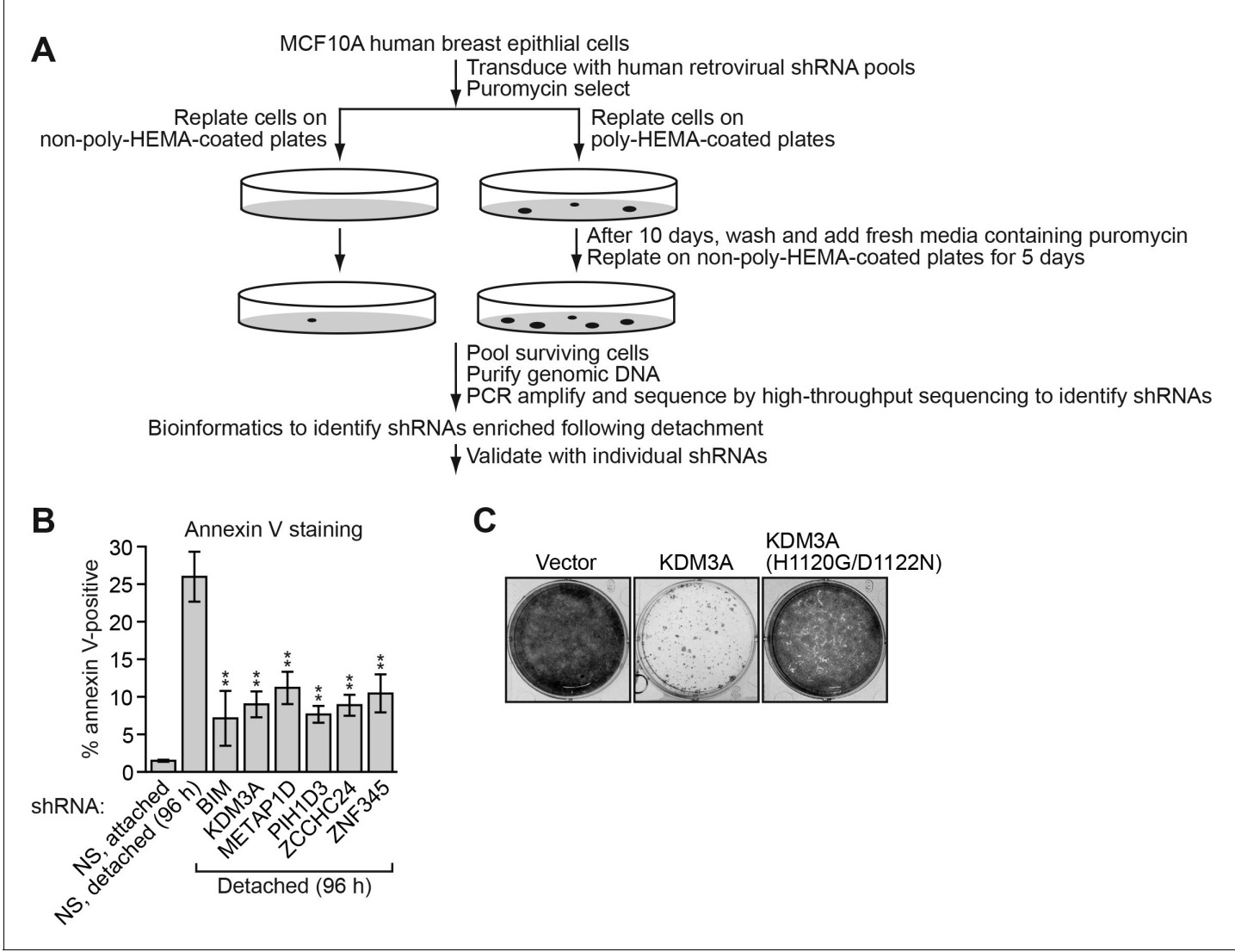

**Figure 1.** Identification of KDM3A as an anoikis effector in breast cancer epithelial cells. (**A**) Schematic of the design of the large-scale RNAi screen to identify anoikis effectors. (**B**) Cell death, monitored by annexin V staining, in MCF10A cells expressing a non-silencing (NS) shRNA and cultured attached to the matrix, or in detached cells (cultured in suspension for 96 hr) expressing a NS shRNA or one of five candidate shRNAs. Error bars indicate SD. *P* value comparisons are made to the detached, NS shRNA control. \*\*p<0.01. (**C**) Crystal violet staining of MCF10A cells expressing vector, KDM3A or the catalytically-inactive KDM3A(H1120G/D1122N) mutant.

The following source data and figure supplements are available for figure 1:

**Source data 1.** List of 26 shRNAs, and the target genes, whose abundance was significantly enriched >500-fold following detachment of MCF10A cells.
**Source data 2.** Source data for *Figure 1B*.
**Figure supplement 1.** FACS analysis.
**Figure supplement 2.** Confirmation of the results of *Figure 1B* using a second, unrelated shRNA.
**Figure supplement 3.** Analysis of *BIM* and candidate shRNA knockdown efficiencies.
**Figure supplement 4.** Confirmation of increased levels of KDM3A upon ectopic expression.

*source data 1*); such shRNAs presumably confer upon MCF10A cells a selective advantage by protecting them from undergoing anoikis.

To validate candidates isolated from the primary screen, we selected the top 20 most highly enriched shRNAs and analyzed them in an independent assay for their ability to confer resistance to anoikis. Briefly, MCF10A cells were transduced with a single shRNA, detached from matrix for 96 hr, and analysed for cell death by annexin V staining. As expected, knockdown of BIM, a positive control, decreased cell death following detachment compared to the control non-silencing (NS) shRNA (*Figure 1B* and *Figure 1—figure supplement 1*). Of the 20 candidate shRNAs tested, five reduced the level of detachment-induced apoptosis compared to the NS shRNA, indicating they conferred anoikis resistance (*Figure 1B* and *Figure 1—figure supplement 1*). Similar results were obtained using a second, unrelated shRNA directed against the same target gene (*Figure 1—figure supplement 2*). Quantitative RT-PCR (qRT-PCR) confirmed in all cases that expression of the target gene was decreased in the knockdown cell line (*Figure 1—figure supplement 3*).

One of the top scoring validated candidates was KDM3A (*Figure 1—source data 1*), a histone demethylase that specifically demethylates mono-methylated (me1) and di-methylated (me2) histone H3 lysine 9 (H3K9) (*Yamane et al., 2006*). H3K9 methylation is a transcriptional repressive mark, and the identification of KDM3A raised the intriguing possibility that induction of anoikis involves transcriptional activation of specific genes through H3K9me1/2 demethylation. Therefore, our subsequent experiments focused on investigating the role of KDM3A in anoikis.

We asked whether ectopic expression of KDM3A was sufficient to promote cell death in attached cells. MCF10A cells were transduced with a retrovirus expressing wild-type KDM3A, a catalytically inactive KDM3A mutant [KDM3A(H1120G/D1122N)] (*Beyer et al., 2008*) or, as a control, empty vector (*Figure 1—figure supplement 4*), and then treated with puromycin for 10 days at which time viability was assessed by crystal violet staining. The results of *Figure 1C* show that ectopic expression of wild-type KDM3A but not KDM3A(H1120G/D1122N) greatly reduced MCF10A cell viability. Collectively, the results of *Figure 1* demonstrate that KDM3A is necessary and sufficient for efficient induction of anoikis in breast epithelial cells.

We next examined the relationship between KDM3A expression and induction of anoikis. The immunoblot of *Figure 2A* shows that KDM3A protein levels were very low in attached MCF10A cells, but robustly increased in a time-dependent manner following detachment. The qRT-PCR analysis of *Figure 2B* shows that an increase in *KDM3A* expression following detachment was also detected at the mRNA level.

We next sought to understand the basis for the increase in KDM3A levels following detachment. As mentioned above, anoikis is suppressed by integrin signaling, which functions through FAK, a regulator of the RAF/MEK/ERK pathway (*Frisch et al., 1996*; *King et al., 1997*). Detachment causes a disruption in integrin–ECM contacts, resulting in a loss of FAK signaling in the detached cells (*Frisch and Francis, 1994*; *Frisch et al., 1996*), which we observed have elevated KDM3A levels (see *Figures 2A and B*). We therefore tested whether restoration of integrin signaling in detached cells would block the increase in KDM3A levels. The results of *Figure 2C* show that the addition of Matrigel basement membrane-like matrix, which restores integrin signaling, to detached cells markedly blocked the elevated levels of the BIM isoform BIM$_{EL}$, as expected, and KDM3A. Treatment of MCF10A cells with a FAK inhibitor increased the levels of KDM3A protein (*Figure 2D*) and mRNA (*Figure 2—figure supplement 1A*). Thus, the increase in KDM3A levels upon detachment of MCF10A cells is due, at least in part, to the loss of integrin/FAK signaling.

We next analyzed the relationship between the EGFR signaling pathway and KDM3A levels. In the first set of experiments, we ectopically expressed either EGFR or a constitutively active MEK mutant, MEK2(S222D/S226D) (MEK2DD) (*Voisin et al., 2008*), both of which have been previously shown to block anoikis in detached cells (*Reginato et al., 2003*). Consistent with these previous results, *Figure 2E* shows that in detached MCF10A cells, expression of either EGFR or MEK2DD substantially decreased the level of BIM$_{EL}$ (*Reginato et al., 2003*). Expression of either EGFR or MEK2DD also decreased the levels of KDM3A in detached MCF10A cells. Conversely, KDM3A protein levels were increased in attached MCF10A cells treated with the EGFR inhibitor gefitinib (*Barker et al., 2001*; *Ward et al., 1994*) (*Figure 2F*) or the MEK inhibitor U0126 (*Favata et al., 1998*) (*Figure 2G*). Both gefitinib and U0126 treatment also resulted in increased *KDM3A* mRNA levels (*Figure 2—figure supplement 1B,C*).

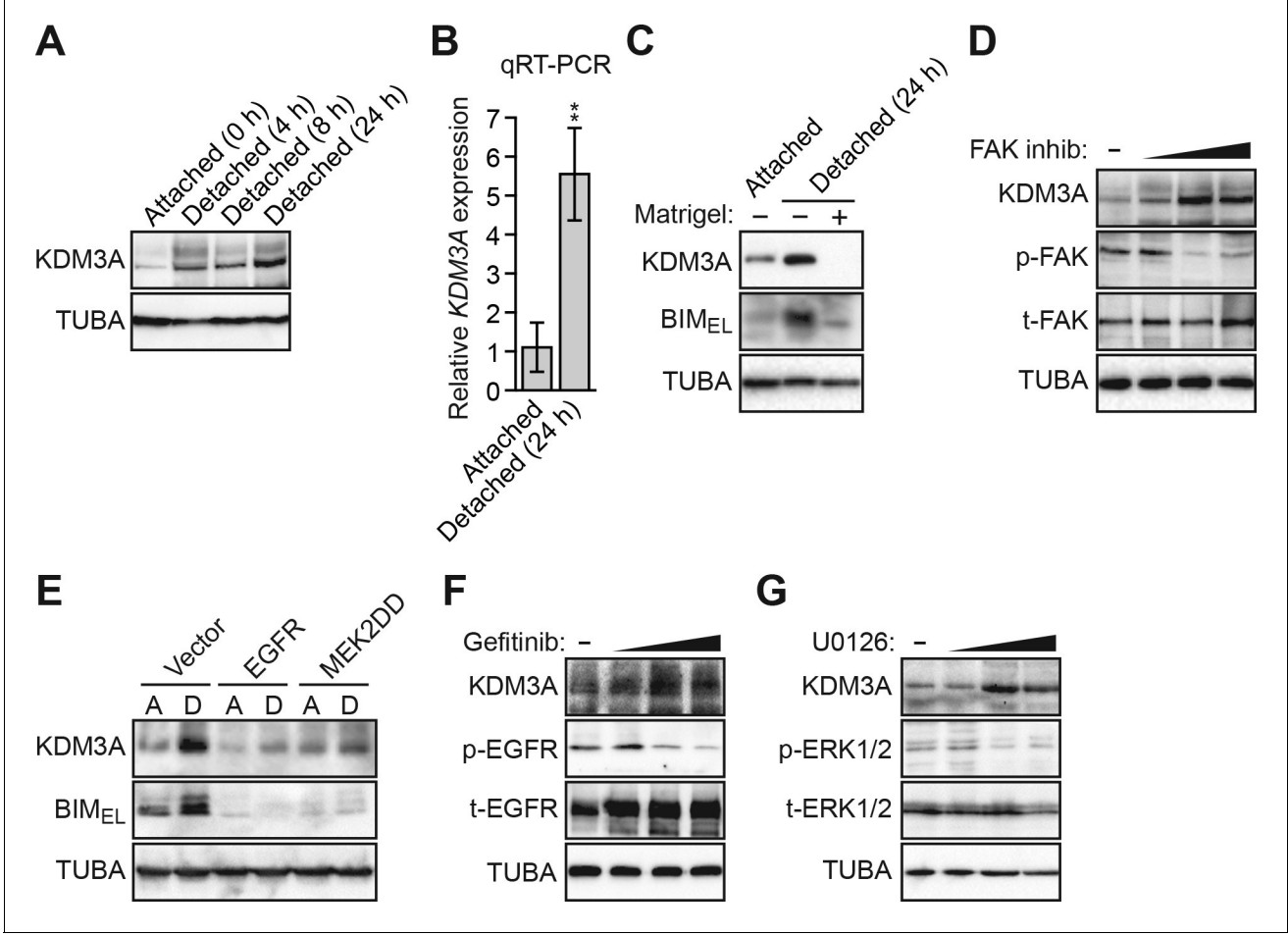

**Figure 2.** Detachment and loss of integrin and growth factor receptor signaling induces KDM3A expression. (**A**) Immunoblot monitoring KDM3A levels in attached MCF10A cells, or detached cells cultured in suspension for 4, 8 or 24 hr. β-actin (ACTB) was monitored as a loading control. (**B**) qRT-PCR analysis monitoring *KDM3A* mRNA levels in attached MCF10A cells, or detached cells cultured in suspension for 24 hr. Error bars indicate SD. **p<0.01. (**C**) Immunoblot monitoring levels of KDM3A and $BIM_{EL}$ in attached MCF10A cells or detached MCF10A cells cultured in suspension for 24 hr and treated in the presence or absence of Matrigel. α-tubulin (TUBA) was monitored as a loading control. (**D**) Immunoblot monitoring levels of KDM3A, phosphorylated FAK (p-FAK) or total FAK (t-FAK) in MCF10A cells treated for 48 hr with 0, 1, 5 or 10 μM FAK inhibitor. (**E**) Immunoblot monitoring levels of KDM3A and $BIM_{EL}$ in MCF10A cells expressing either vector, EGFR or MEK2DD and cultured as attached (A) or detached (D) cells grown in suspension for 24 hr. (**F**) Immunoblot monitoring levels of KDM3A, phosphorylated EGFR (p-EGFR) or total EGFR (t-EGFR) in MCF10A cells treated for 48 hr with 0, 1, 5 or 10 μM gefitinib. (**G**) Immunoblot monitoring levels of KDM3A, phosphorylated ERK1/2 (p-ERK1/2) or total ERK1/2 (t-ERK1/2) in MCF10A cells treated for 48 hr with 0, 1, 5 or 10 μM U0126.

The following source data and figure supplement are available for figure 2:

**Source data 1.** Source data for *Figure 2B*.

**Figure supplement 1.** Inhibition of FAK, EGFR, or MEK in MCF10A cells increases *KDM3A* expression.

The results described above suggest a model in which following detachment, the resulting increase in KDM3A demethylates H3K9me1/2 to stimulate expression of one or more pro-apoptotic genes. To test this model and identify pro-apoptotic KDM3A target genes, we took a candidate-based approach and analyzed expression of a panel of genes encoding pro-apoptotic BCL2 proteins (*Boyd et al., 1994*; *Lomonosova and Chinnadurai, 2008*; *Matsushima et al., 1998*) in attached MCF10A cells and detached cells expressing a NS or *KDM3A* shRNA. We sought to identify genes whose expression increased following detachment in control but not in *KDM3A* knockdown cells. We found that expression of the vast majority of genes encoding pro-apoptotic BCL2 proteins were

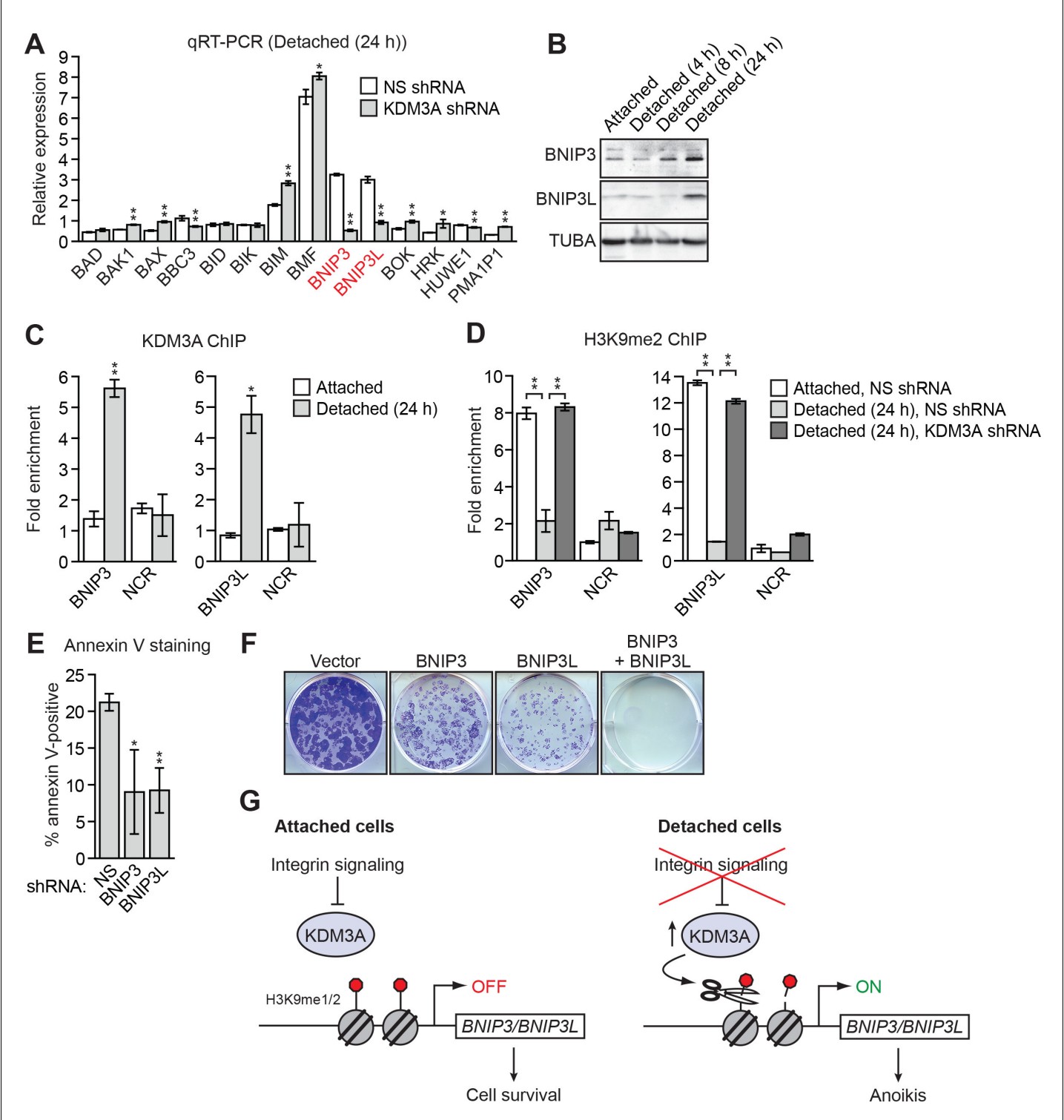

**Figure 3.** KDM3A induces anoikis by transcriptionally activating *BNIP3* and *BNIP3L*. (**A**) qRT-PCR analysis monitoring expression of pro-apoptotic BCL2 genes in detached MCF10A cells grown in suspension for 24 hr and expressing a NS or *KDM3A* shRNA. The expression of each gene is shown relative to that obtained in attached cells expressing a NS shRNA, which was set to 1. *P* value comparisons for each gene are made to the NS shRNA control. Genes whose expression is decreased >2-fold upon *KDM3A* knockdown are indicated in red. (**B**) Immunoblot analysis monitoring levels of BNIP3 and BNIP3L in attached MCF10A cells, and detached cells following growth in suspension for 4, 8 or 24 hr. (**C**) ChIP monitoring binding of KDM3A on the promoters of *BNIP3* and *BNIP3L* or a negative control region (NCR) in attached MCF10A cells or detached cells grown in suspension for 24 hr. *P* value comparisons for each region are made to the attached control. (**D**) ChIP monitoring the levels of H3K9me2 on the promoters of *BNIP3* and *BNIP3L* or a

*Figure 3 continued on next page*

*Figure 3 continued*

negative control region in attached MCF10A cells or detached cells expressing a NS or *KDM3A* shRNA and grown in suspension for 24 hr. *P* value comparisons for each region are made to the detached, NS shRNA control. (**E**) Cell death, monitored by annexin V staining, in MCF10A cells expressing a NS, *BNIP3* or *BNIP3L* shRNA. (**F**) Crystal violet staining of MCF10A cells expressing vector, BNIP3, BNIP3L or both BNIP3 and BNIP3L. (**G**) Model. Error bars indicate SD. *p<0.05; **p<0.01.

The following source data and figure supplements are available for figure 3:

**Source data 1.** Source data for *Figure 3A, C, D and E*.
**Figure supplement 1.** Confirmation of the results of *Figure 3A* using a second, unrelated *KDM3A* shRNA.
**Figure supplement 2.** The level of H3K9me1 on the *BNIP3* and *BNIP3L* promoters is diminished following detachment, which is counteracted by knockdown of *KDM3A*.
**Figure supplement 3.** Overexpression of KDM3A, but not KDM3A(H1120G/D1122N), in attached MCF10A cells results in decreased levels of H3K9me1 and H3K9me2 on the *BNIP3* and *BNIP3L* promoters and increased expression of *BNIP3* and *BNIP3L*.
**Figure supplement 4.** Analysis of *BNIP3* and *BNIP3L* shRNA knockdown efficiencies.
**Figure supplement 5.** Confirmation of the results of *Figure 3E* using a second, unrelated shRNA.
**Figure supplement 6.** Confirmation of increased levels of BNIP3 and BNIP3L upon ectopic expression.

unaffected by detachment in MCF10A cells (*Figure 3A* and *Figure 3—figure supplement 1*). Consistent with previous results (*Reginato et al., 2003*; *Schmelzle et al., 2007*), expression of *BIM* and *BMF* were increased upon detachment. However, knockdown of *KDM3A* did not decrease expression of either *BIM* or *BMF*. By contrast, following detachment, expression of *BNIP3* and *BNIP3L* increased, and were the only genes whose expression was diminished more than 2-fold by *KDM3A* knockdown (*Figure 3A* and *Figure 3—figure supplement 1*). We therefore performed a series of experiments to determine whether *BNIP3* and *BNIP3L* are critical KDM3A target genes that mediate anoikis.

In the first set of experiments we analyzed BNIP3 and BNIP3L protein levels during anoikis induction. The immunoblot of *Figure 3B* shows that BNIP3 and BNIP3L levels were very low in attached cells and substantially increased following detachment, with a time course similar to that of detachment-induced KDM3A expression (see *Figure 2A*). The chromatin immunoprecipitation (ChIP) experiment of *Figure 3C* shows that KDM3A was bound to the *BNIP3* and *BNIP3L* promoters in detached but not attached cells. Moreover, the levels of H3K9me2 (*Figure 3D*) and H3K9me1 (*Figure 3—figure supplement 2*) on the *BNIP3* and *BNIP3L* promoters were greatly diminished following detachment, which was counteracted by knockdown of *KDM3A*. Conversely, overexpression of KDM3A but not KDM3A(H1120G/D1122N) in attached MCF10A cells resulted in decreased levels of H3K9me1 and H3K9me2 on the *BNIP3* and *BNIP3L* promoters and increased expression of *BNIP3* and *BNIP3L* (*Figure 3—figure supplement 3*). Finally, knockdown of *BNIP3* or *BNIP3L* (*Figure 3—figure supplement 4*) resulted in decreased apoptosis following detachment (*Figure 3E* and *Figure 3—figure supplement 5*). To further establish the pro-apoptotic role of BNIP3 and BNIP3L in MCF10A cells, we ectopically expressed BNIP3, BNIP3L or both in attached cells (*Figure 3—figure supplement 6*). *Figure 3F* shows that moderate cell death was observed upon ectopic expression of either BNIP3 or BNIP3L, but substantial cell death occurred in cells ectopically expressing both BNIP3 and BNIP3L. Collectively, these results establish *BNIP3* and *BNIP3L* as critical KDM3A target genes that mediate anoikis (*Figure 3G*).

We considered the possibility that decreased *KDM3A* expression may contribute to anoikis resistance in breast cancer cells and performed a series of experiments to test this idea. We first analyzed a panel of human breast cancer cell lines (BT549, MDA-MB-231, MCF7, SUM149 and T47D) comparing, as a control, anoikis-sensitive MCF10A cells. As expected, detachment-induced apoptosis was significantly diminished in breast cancer cell lines compared to MCF10A cells, indicative of anoikis

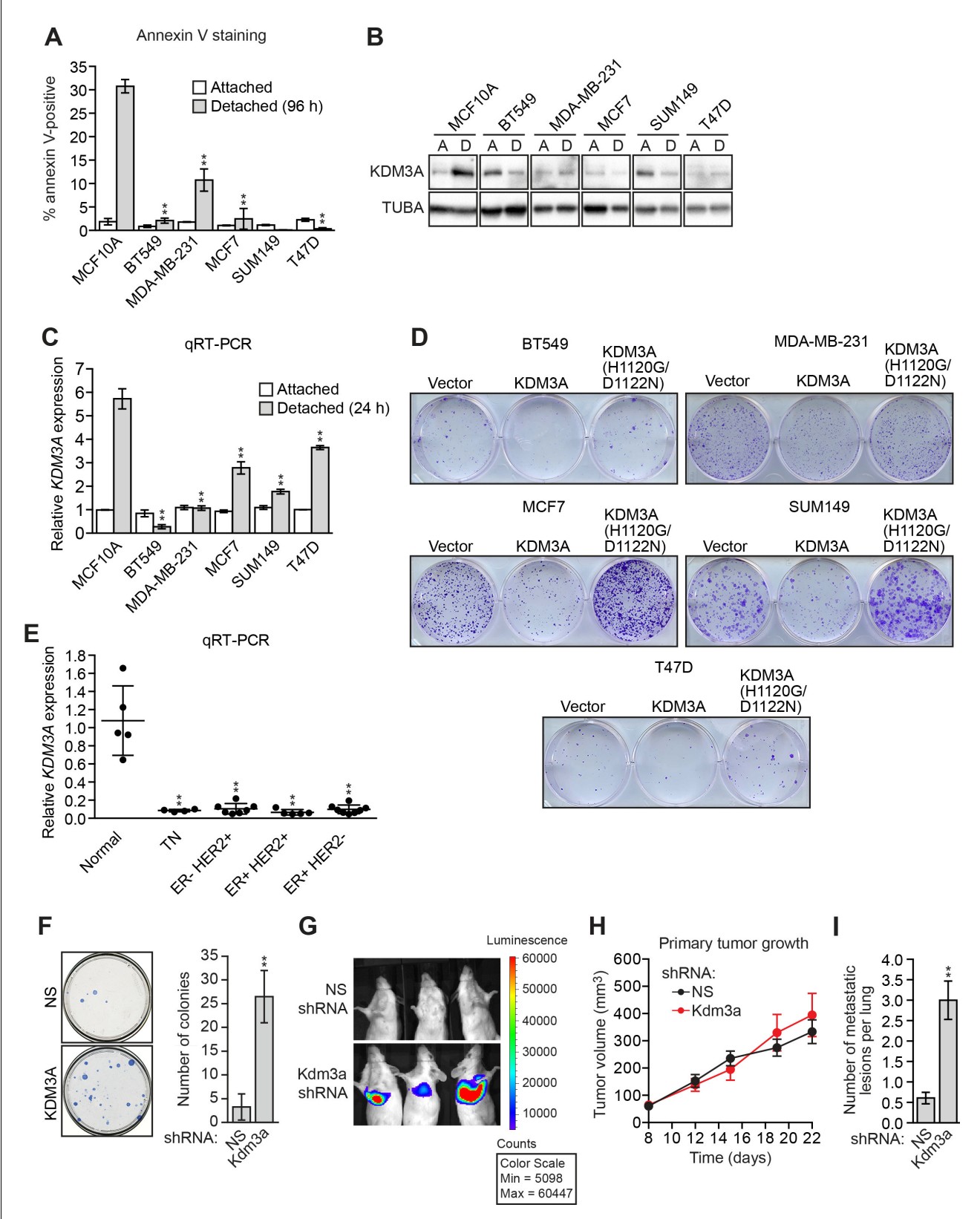

**Figure 4.** *KDM3A* prevents metastasis and its expression is defective in human breast cancer cell lines and tumors. (**A**) Cell death, monitored by annexin V staining, in MCF10A cells and a panel of human breast cancer cell lines cultured as attached cells or detached following growth in

*Figure 4 continued on next page*

*Figure 4 continued*

suspension for 96 hr. Error bars indicate SD. *P* value comparisons for each breast cancer cell line are made to the detached MCF10A sample. (**B**) Immunoblot analysis monitoring KDM3A levels in MCF10A cells and a panel of human breast cancer cell lines cultured as attached (A) cells or detached (D) following growth in suspension for 24 hr. All images for the KDM3A antibody were cropped from the same blot and thus were processed and exposed in the same manner, as were images for the TUBA loading control. (**C**) qRT-PCR analysis monitoring *KDM3A* expression in MCF10A cells and a panel of human breast cancer cell lines cultured as attached cells or detached following growth in suspension for 24 hr. Error bars indicate SD. *P* value comparisons for each breast cancer cell line are made to the detached MCF10A sample. (**D**) Crystal violet staining of human breast cancer cells expressing vector, KDM3A or KDM3A(H1120G/D1122N). (**E**) qRT-PCR analysis monitoring *KDM3A* expression in normal breast epithelial cells and human breast tumors. TN, triple negative [estrogen receptor-negative (ER-), human epidermal growth factor receptor 2-negative (HER2-) and progesterone receptor-negative (PR-)]. Error bars indicate SD. The differences in *KDM3A* expression between subtypes are not statistically significant. (**F**) Mouse pulmonary survival assay. (Left) Representative plates showing colony formation of CLS1 cells expressing a NS or *Kdm3a* shRNA that had been isolated from mouse lungs following tail vein injection. (Right) Quantification of colony formation (n = 4 mice per shRNA). Error bars indicate SD. (**G**) Live animal imaging monitoring lung tumor metastasis in mice following injection of 67NR cells expressing a NS or *Kdm3a* shRNA (n = 3 mice per group). (**H**) Primary tumor growth in mice injected with 4T07 cells expressing a NS (n = 7) or *Kdm3a* (n = 8) shRNA. Error bars indicate SEM. The differences in primary tumor growth between groups are not statistically significant. (**I**) Metastatic burden. Number of metastatic lesions per lung in mice injected with 4T07 cells expressing a NS (n = 7) or *Kdm3a* (n = 8) shRNA. Error bars indicate SEM. **p<0.01.

The following source data and figure supplements are available for figure 4:

**Source data 1.** Source data for *Figure 4A, C, E, F, H and I*.

**Figure supplement 1.** FACS analysis.

**Figure supplement 2.** Oncomine analysis of *KDM3A* expression in breast cancer.

**Figure supplement 3.** Analysis of basal *KDM3A* expression in human breast cancer cell lines.

**Figure supplement 4.** Analysis of *Kdm3a* shRNA knockdown efficiency in mouse CLS1 cells.

**Figure supplement 5.** Analysis of *Kdm3a* expression in a mouse breast cancer carcinoma progression series.

**Figure supplement 6.** Analysis of *Kdm3a* shRNA knockdown efficiency in mouse 67NR cells.

**Figure supplement 7.** Analysis of *Kdm3a* shRNA knockdown efficiency in mouse 4T07 cells.

**Figure supplement 8.** Confirmation of the results of *Figure 4H* using a second, unrelated shRNA.

**Figure supplement 9.** Confirmation of the results of *Figure 4I* using a second, unrelated shRNA.

resistance (*Figure 4A* and *Figure 4—figure supplement 1*). Moreover, following detachment of the breast cancer cell lines, induction of KDM3A at both the protein (*Figure 4B*) and mRNA (*Figure 4C*) levels was much lower than that observed in MCF10A cells. However, ectopic expression of KDM3A was sufficient to induce apoptosis in each of the five breast cancer cell lines (*Figure 4D*). Collectively, these results indicate that anoikis-resistance of human breast cancer cells is due, at least in part, to inefficient induction of KDM3A following detachment.

We next analyzed *KDM3A* expression in human breast cancer patient samples. Interrogation of the Oncomine database (*Rhodes et al., 2007*) revealed decreased expression levels of *KDM3A* in several breast cancer datasets (*Figure 4—figure supplement 2*). To confirm these in silico results, we analyzed *KDM3A* expression by qRT-PCR in a series of human breast cancer patient samples. The results of *Figure 4E* show that compared to normal breast epithelium *KDM3A* expression was significantly decreased in a high percentage of breast cancers. Likewise, basal *KDM3A* expression levels were also diminished in most human breast cancer cell lines analyzed (*Figure 4—figure supplement 3*).

Finally, we performed a series of experiments to determine whether KDM3A affects metastatic potential. We first asked whether depletion of KDM3A would promote anoikis resistance in vivo using a mouse pulmonary survival assay. Briefly, immortalized but non-transformed mouse mammary

epithelial CLS1 cells were stably transduced with an NS or *Kdm3a* shRNA (*Figure 4—figure supplement 4*) and injected into the tail vein of syngeneic mice. After 2 weeks, the lungs were harvested, dissociated into single cell suspensions, and plated in media containing puromycin to select for cells expressing the shRNA. The surviving colonies were visualized by crystal violet staining and quantified. The results of *Figure 4F* show that *Kdm3a* knockdown significantly increased the number of cells that survived in the mouse lung relative to the control NS shRNA.

In a second set of experiments, we used a well-characterized mouse breast cancer carcinoma progression series comprising isogenic cell lines with increasing metastatic potential: (1) non-invasive and non-metastatic 67NR cells, which form primary tumors, (2) invasive and non-metastatic 4T07 cells, which enter the circulation but fail to establish secondary tumors, and (3) highly metastatic 4T1 cells, which disseminate widely and colonize distant organ sites (*Aslakson and Miller, 1992*). qRT-PCR analysis revealed decreased *Kdm3a* expression in cell lines with greater metastatic potential (*Figure 4—figure supplement 5*). We expressed either a control NS shRNA or a *Kdm3a* shRNA in 67NR cells containing a luciferase reporter gene (*Figure 4—figure supplement 6*). Cells were injected into the tail veins of three syngeneic mice and pulmonary metastases were visualized by live animal imaging after 5 weeks. The results of *Figure 4G* show, as expected, that control 67NR cells failed to form pulmonary metastases in any of the three mice analyzed. By contrast, *Kdm3a* knockdown 67NR cells formed substantial pulmonary metastases in all three mice.

Finally, in a more stringent metastasis experiment, control and *Kdm3a* knockdown 4T07 cells (*Figure 4—figure supplement 7*), a non-metastatic mouse breast cancer cell line, were injected in the mammary fat pad of ten syngeneic mice. After 22 days the primary tumors were surgically removed and 8 weeks post-injection the animals were sacrificed and pulmonary tumors quantified. The growth of primary tumors formed by NS or *Kdm3a* knockdown cells was similar (*Figure 4H* and *Figure 4—figure supplement 8*). However, *Kdm3a* knockdown cells caused significantly increased metastatic burden in the lungs compared to control 4T07 cells (*Figure 4I* and *Figure 4—figure supplement 9*). Consistent with our results, knockdown of *Bnip3* has also been shown to cause increased metastasis in similar in vivo experiments (*Manka et al., 2005*). Collectively, these results show that KDM3A functions to prevent metastasis.

Based on the results presented above, we propose a model of anoikis induction that is illustrated in *Figure 3G* and discussed below. Following detachment of non-transformed cells, integrin signaling is decreased leading to transcriptional induction of *KDM3A*. The increased levels of KDM3A results in its recruitment to the pro-apoptotic genes *BNIP3* and *BNIP3L,* where it promotes demethylation of inhibitory H3K9me1/2 marks and transcriptional activation of the two genes, resulting in anoikis induction. Consistent with this model, previous studies have shown that hypoxia results in transcriptional activation of *KDM3A, BNIP3* and *BNIP3L* (*Beyer et al., 2008*; *Sowter et al., 2001*). We have found that in anoikis-resistant human breast cancer cell lines and tumors, *KDM3A* expression is defective, highlighting the importance of this pathway in promoting anoikis. Collectively, our results reveal a novel transcriptional regulatory program that mediates anoikis in non-transformed cells and is disabled during cancer development.

As described above, previous studies have shown that BIM and BMF are also effectors of anoikis (*Reginato et al., 2003*; *Schmelzle et al., 2007*). However, we have found that unlike BNIP3 and BNIP3L, BIM and BMF are not regulated by KDM3A. Thus, our results reveal that anoikis is promoted by multiple non-redundant pathways, which may help prevent the development of anoikis resistance.

## Materials and methods

### Cell lines and culture

T47D, MDA-MB-231, BT549 and CLS1 cells were obtained from ATCC (Manassas, VA) and grown as recommended by the supplier. MCF7 cells (National Cancer Institute, Bethesda, MD) were maintained in DMEM (GE Healthcare Life Sciences, Marlborough, MA) supplemented with 1X nonessential amino acids (NEAA; Thermo Scientific, Waltham, MA) and 10% fetal bovine serum (FBS; Atlanta Biologics, Norcross, GA). MCF10A cells (ATCC) were maintained in DMEM/F12 (GE Healthcare Life Sciences) supplemented with 5% donor horse serum (Thermo Scientific), 20 ng/ml epidermal growth factor (Peprotech, Rocky Hill, NJ), 10 μg/ml insulin (Life Technologies, Grand Island, NY), 1 ng/ml

cholera toxin (Sigma-Aldrich, St. Louis, MO), 100 µg/ml hydrocortisone (Sigma-Aldrich), 50 U/ml penicillin (Thermo Scientific), and 50 µg/ml streptomycin (Invitrogen, Grand Island, NY). SUM149 cells were obtained from Dr. Donald Hnatowich (University of Massachusetts Medical School, Worcester, MA) and grown in RPMI (Invitrogen) supplemented with 10% FBS, 0.01% insulin, 50 U/ml penicillin, and 50 µg/ml streptomycin. 67NR and 4T07 cells were obtained from Dr. Fred Miller (Wayne State University School of Medicine, Detroit, MI) and were grown in high glucose DMEM (GE Healthcare Life Sciences) supplemented with 10% FBS, 50 U/ml penicillin, and 50 µg/ml streptomycin. Cell lines used in this study have not been authenticated for identity.

## Ectopic expression

*KDM3A* and *KDM3A(H1120G/D1122N)* were PCR amplified from pCMV-JMJD1A and pCMV-JMJD1A(H1120G/D1122N), respectively, obtained from Dr. Peter Staller (Biotech Research and Innovation Centre, University of Copenhagen, Denmark), using primers (forward, 5′-CTCGAGCCG TTAAGGTTTGCCAAAAC-3′ and reverse, 5′-ATCGTTAACAGGGAGATTAAGGTTTGCCA-3′) engineered with XhoI and HpaI restriction sites and then cloned into pMSCVpuro (ClonTech Laboratories, Inc., Mountain View, CA). *BNIP3L* was PCR amplified from Bnip3L pcDNA3.1 (plasmid #17467, Addgene, Cambridge, MA) using primers (forward, 5′-AATCTCGAGCATGTCGTCCCACCTAGT-3′ and reverse 5′-ATCGAATTCTTAATAGGTGCTGGCAGAGG-3′) engineered with XhoI and EcoRI restriction sites and cloned into pMSCVhygro (ClonTech Laboratories, Inc.). *BNIP3* was PCR amplified from MGC Human *BNIP3* cDNA (Dharmacon, Marlborough, MA) using primers (forward, 5′-AA TCTCGAGCATGTCGCAGAACGGAGCG-3′ and reverse 5′-ATCGAATTCACTAAATTAGGAACG-CAGCAT-3′) engineered with XhoI and EcoRI restriction sites and cloned into pMSCVpuro.

MCF10A cells stably expressing pMSCVpuro-JMJD1A, pMSCVpuro-JMJD1A-H1120G/D1122N, pMSCVpuro-BNIP3, pMSCVhygro-BNIP3L, pMSCVpuro-empty, pMSCVhygro-empty, pBABE-MEK2DD (obtained from Dr. Sylvain Meloche, Université de Montréal), pBABE-EGFR (Addgene), or pBABE-empty (Addgene) were generated by retroviral transduction as described previously (*Santra et al., 2009*). Twelve days after puromycin or hygromycin selection, cells were stained with 0.5% crystal violet.

## RNA interference

The human shRNA^mir pSM2 library (Open Biosystems/Thermo Scientific, Pittsburgh, PA) was obtained through the University of Massachusetts Medical School RNAi Core Facility (Worcester, MA). Retroviral pools were generated and used to transduce MCF10A cells as described previously (*Gazin et al., 2007*). Following puromycin selection, transduced cells were divided into two populations: one was plated on poly-HEMA-coated tissue culture plates (plates were coated with poly-HEMA (20 mg/ml) (Sigma-Aldrich), dried at room temperature overnight, and washed with phosphate buffered saline (PBS) before use) and grown for 10 days, and the other was grown for 10 days under normal tissue culture conditions. Cells that survived 10 days in suspension (a time point at which >95% of cells transduced with the control NS shRNA were killed) were seeded under normal tissue culture conditions to expand the population. shRNAs present in the surviving suspension population and the attached population were identified by deep sequencing at the University of Massachusetts Medical School Deep Sequencing Core Facility (Worcester, MA). The frequency of individual shRNAs in each sample was determined as described previously (*Xie et al., 2012*). The raw sequencing data have been uploaded to NCBI Gene Expression Omnibus and are accessible through GEO Series accession number GSE80144.

For stable shRNA knockdowns, $1 \times 10^5$ cells were seeded in a six-well plate to 50% confluency and subsequently transduced with 200 µl lentiviral particles expressing shRNAs (obtained from Open Biosystems/Thermo Scientific through the UMMS RNAi Core Facility, listed in *Supplementary file 1*) in a total volume of 1 ml of appropriate media supplemented with 6 µg/ml polybrene (Sigma-Aldrich). Media was replaced after overnight incubation to remove the polybrene, and viral particles and cells were subjected to puromycin selection (2 µg/ml) for 3 days.

## qRT-PCR

Total RNA was isolated and reverse transcription was performed as described (*Gazin et al., 2007*), followed by qRT-PCR using Power SYBR Green PCR Master Mix (Applied Biosystems, Grand Island,

NY). *RPL41* or *GAPDH* were used as internal reference genes for normalization. See *Supplementary file 2* for primer sequences. Each sample was analyzed three independent times and the results from one representative experiment, with technical triplicates or quadruplicates, are shown.

### Anoikis assays

Cells were placed in suspension in normal growth media in the presence of 0.5% methyl cellulose (Sigma Aldrich) (to avoid clumping of cells) on poly-HEMA-coated tissue culture plates. All anoikis assays were done at a cell density of $3 \times 10^5$ cells/ml. Control cells were cultured under normal cell culture conditions. Cell death was measured by staining the cells with FITC-conjugated Annexin-V (ApoAlert, ClonTech) according to the manufacturer's instructions followed by analysis by flow cytometry (Flow Cytometry Core Facility, University of Massachusetts Medical School) at the indicated times. To restore integrin signaling in suspension, media was supplemented with 5% growth-factor-reduced Matrigel (BD Biosciences, San Diego, CA). Each sample was analyzed in biological triplicate.

### Immunoblot analysis

Cell extracts were prepared by lysis in Laemmli buffer in the presence of protease inhibitor cocktail (Roche, Indianapolis, IN). The following commercial antibodies were used: beta-ACTIN (Sigma-Aldrich); BNIP3, BNIP3L, KDM3A, H3K9me2 (all from Abcam, Cambridge, MA); cleaved Caspase 3, BIM, phospho-ERK1/2, total ERK1/2, phospho-EGFR, total EGFR, phospho-FAK (all from Cell Signaling Technology, Danvers, MA); total FAK (Millipore, Billerica, MA); and α-tubulin (TUBA; Sigma-Aldrich).

### Chemical inhibitor treatment

Cells were treated with dimethyl sulfoxide (DMSO), 1, 5 or 10 µM U0126 (Cell Signaling Technology), gefitinib (Santa Cruz Biotechnology, Inc., Dallas, TX), or FAK inhibitor 14 (CAS 4506-66-5, Santa Cruz Biotechnology, Inc.) for 48 hr prior to preparation of cell extracts or total RNA isolation, as described above.

### ChIP assays

ChIP assays were performed as previously described (*Gazin et al., 2007*) using antibodies against KDM3A and H3K9me2 (both from Abcam) and H3K9me1 (Epigentek). ChIP products were analyzed by qPCR (see *Supplementary file 2* for promoter-specific primer sequences). Samples were quantified as percentage of input, and then normalized to an irrelevant region in the genome (~3.2 kb upstream from the transcription start site of GCLC). Fold enrichment was calculated by setting the IgG control IP sample to a value of 1. Each ChIP experiment was performed three independent times and the results from one representative experiment, with technical duplicates, are shown.

### Analysis of *KDM3A* expression in human breast cancer samples

This study was approved by the institutional review boards at the University of Massachusetts Medical School (UMMS) and the Mayo Clinic. Total RNA from 24 breast cancer patient samples were obtained from Fergus Couch (Mayo Clinic, Rochester, MN) and total RNA from five normal breast samples were obtained from the University of Massachusetts Medical School Tissue and Tumor Bank Facility. *KDM3A* expression was measured by qRT-PCR in technical triplicates of each patient sample. Statistical analysis (unequal variance t-test) was performed using R, a system for statistical computation and graphics (*Ihaka and Gentleman, 1996*). The Oncomine Cancer Profiling Database (Compendia Bioscience, Ann Arbor, MI) was queried using the cancer type Breast Cancer and a threshold p-value of 0.05 to access Finak (*Finak et al., 2008*), Sorlie (*Sorlie et al., 2001*), Zhao (*Zhao et al., 2004*) and TCGA (*TCGA, 2011*) datasets. Histograms depicting *KDM3A* gene expression in each sample, and the p value for the comparison of *KDM3A* expression between the groups, were obtained directly through the Oncomine software.

## Animal experiments

All animal protocols were approved by the Institution Animal Care and Use Committee (IACUC). Animal sample sizes were selected based on precedent established from previous publications.

## In vivo anoikis assays

CLS1 cells were stably transduced with either a NS or *Kdm3a* shRNA and selected with 2 µg/ml puromycin for 5 days. Stably transduced CLS1 cells ($2 \times 10^5$) were injected into the tail vein of 4–6 week old female BALB/c mice (Taconic Biosciences) (n = 4 mice per shRNA). Two weeks post injection the lungs were harvested, dissociated into single cell suspension, and plated onto tissue culture plates. Transduced CLS1 cells were selected for by treating the dissociated lung cells with 2 µg/ml puromycin. Surviving colonies were stained with crystal violet and quantified by counting. All experiments were performed in accordance with the Institutional Animal Care and Use Committee (IACUC) guidelines.

## Pulmonary tumor assay

67NR cells were transduced with a NS or *Kdm3a* shRNA and selected with 2 µg/ml puromycin for 5 days. Stably transduced 67NR cells ($2 \times 10^5$) were injected into the tail vein of 6–8 week old female BALB/c mice (n = 3 mice per shRNA). Five weeks post injection, mice were given an intraperitoneal injection of D-Luciferin (100 mg/kg) (Gold Biotechnology, St. Louis, MO) and imaged on the Xenogen IVIS-100 (Caliper Life Sciences). Images were taken with Living Image software. All experiments were performed in accordance with the Institutional Animal Care and Use Committee (IACUC) guidelines.

## Spontaneous metastasis assays

Female BALB/c mice (4–6 weeks) were purchased from Charles River Laboratories (Shrewsbury, MA). The mice were housed in facilities managed by the McGill University Animal Resources Centre (Montreal, Canada), and all animal experiments were conducted under a McGill University–approved Animal Use Protocol in accordance with guidelines established by the Canadian Council on Animal Care.

Spontaneous metastasis studies were carried out as previously described (*Tabaries et al., 2011*). Briefly, 4T07 cells expressing a NS or *Kdm3a* shRNA were first tested for mycoplasma contamination and found to be negative. Cells were then harvested from subconfluent plates, washed once with PBS, and resuspended ($5 \times 10^3$ cells) in 50 µl of a 50:50 solution of Matrigel (BD Biosciences) and PBS. This cell suspension was injected into the right abdominal mammary fat pad of BALB/c mice (n = 10 mice per shRNA) and measurements were taken beginning on day 7 post-injection. Animals that did not develop a primary tumor were excluded from the study. Tumor volumes were calculated using the following formula: $\pi LW^2/6$, where $L$ is the length and $W$ is the width of the tumor. Tumors were surgically removed, using a cautery unit, once they reached a volume around 500 mm$^3$, approximately 3 weeks post injection. Lungs were collected 8 weeks post-injection. Tumor burden in the lungs was quantified from four H&E stained step sections (200 µm/step). The number of lesions per section were counted using Imagescope software (Aperio, Vista, CA).

## Statistics

All quantitative data were collected from experiments performed in at least triplicate, and expressed as mean ± standard deviation, with the exception of *Figure 4H and I*, which are expressed as mean ± SEM. Differences between groups were assayed using two-tailed Student's t test, except where noted above. Significant differences were considered when P<0.05.

## Acknowledgements

We thank Fred Miller, Donald Hnatowich, Peter Staller, Sylvain Meloche, Fergus Couch for reagents; Douglas Green for insightful suggestions; the UMMS RNAi Core Facility for providing for shRNA clones; Lynn Chamberlain and Alysia R Bryll for experimental assistance; and Sara Deibler for editorial assistance. This work was supported by a Department of Defense Breast Cancer Research

Program grant (BC060871) and a US National Institutes of Health grant (R01GM033977) to MRG., who is also an investigator of the Howard Hughes Medical Institute.

## Additional information

### Funding

| Funder | Grant reference number | Author |
|---|---|---|
| National Institutes of Health | R01GM033977 | Michael R Green |
| Howard Hughes Medical Institute | 068101 | Michael R Green |
| U.S. Department of Defense | BC060871 | Michael R Green |

The funders had no role in study design, data collection and interpretation, or the decision to submit the work for publication.

### Author contributions

VEP, SG, Conception and design, Acquisition of data, Analysis and interpretation of data, Drafting or revising the article; ST, TMS, PMS, Acquisition of data, Analysis and interpretation of data, Drafting or revising the article; LJZ, Analysis and interpretation of data, Drafting or revising the article; MRG, Conception and design, Analysis and interpretation of data, Drafting or revising the article

### Author ORCIDs

Michael R Green, http://orcid.org/0000-0003-3017-3298

### Ethics

Human subjects: This study was approved by the institutional review boards at the University of Massachusetts Medical School (UMMS) and the Mayo Clinic. Consent to publish is not necessary as all patient samples were de-identified.

Animal experimentation: This study was performed in strict accordance with the recommendations in the Guide for the Care and Use of Laboratory Animals of the National Institutes of Health. All of the animals were handled according to approved institutional animal care and use committee (IACUC) protocols of the University of Massachusetts Medical School (A-2300) and McGill University. All surgery was performed under sodium pentobarbital anesthesia, and every effort was made to minimize suffering.

## Additional files

### Supplementary files

• Supplementary file 1. List of shRNAs obtained from Open Biosystems/Thermo Scientific.
• Supplementary file 2. List of primers used for qRT-PCR and ChIP.

### Major datasets

The following dataset was generated:

| Author(s) | Year | Dataset title | Dataset URL | Database, license, and accessibility information |
|---|---|---|---|---|
| Pedanou VE, Green MR | 2016 | The histone H3K9 demethylase KDM3A promotes anoikis through transcriptional activation of pro-apoptotic genes BNIP3 and BNIP3L | http://www.ncbi.nlm.nih.gov/geo/query/acc.cgi?acc=GSE80144 | Publicly available at the NCBI Gene Expression Omnibus (accession no. GSE80144) |

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
