## [Decision Letter]

Thank you for submitting your article "The histone H3K9 demethylase KDM3A promotes anoikis by transcriptionally activating pro-apoptotic genes *BNIP3* and *BNIP3L*" for consideration by *eLife*. Your article has been reviewed by two peer reviewers, and the evaluation has been overseen by Ali Shilatifard as the Reviewing Editor and James Manley as the Senior Editor. The two reviewers have opted to remain anonymous.

The reviewers have discussed the reviews with one another and the Reviewing Editor and there is a clear agreement that your study is very well done and the manuscript is a suitable candidate for publication in *eLife* following the implementation of the minor experimental suggestions and the textual modifications indicated in the reviewers' comments.

*Reviewer #1:*

In their short report Green and colleagues identify the histone H3K9me2/me1 demethylase KDM3A in an unbiased shRNA screen as a positive regulator of anoikis, a form of apoptosis that is initiated upon cell detachment from the extracellular matrix. KDM3A is transcriptionally induced upon cell detachment as a result of reduced integrin signaling through focal adhesion kinase (FAK) and the EGFR pathway via RAF/MEK/ERK and demethylates H3K9me2 on the promoters of the proapoptotic genes BNIP3 and BNIP3L resulting in their transcriptional induction and apoptosis. Further experiments provide clear evidence that KDM3A mRNA levels are generally kept lower in metastatic cell lines/tumors versus non-metastatic cell lines/tumors suggesting a mechanism by which detached cancer cells can avoid apoptosis and invade other tissues.

What makes this paper particularly strong is the fact that it starts with a functional phenotypic assay (anoikis) and an unbiased shRNA screen so that the identification of KDM3A as the second strongest hit makes it a very promising candidate from the very start. The manuscript is written very clearly, logically and is easy to follow (not convoluted which is a great plus). The experiments seem to carefully build on each other and delineate a clear mechanistic pathway from the beginning of cell detachment to apoptosis. Furthermore, Figure 4 provides relevant data highlighting the importance of the KDM3A/anoikis axis for the clinical setting. In summary, this manuscript not only introduces a novelty aspect by implicating KDM3A in anoikis but also provides the mechanistic and clinical framework to make this an appropriate story for *eLife*.

However, some details are missing that should be addressed:

1) Is the FAK and EFGR pathway also required to regulate KDM3A on the transcriptional level? Figure 2 only provides evidence that this is the case on the protein level.

2) Discrepancy in KDM3A levels between Figure 2 and other westerns throughout the paper. KDM3A levels appear to be virtually absent under attached conditions in Figure 2 whereas generally the difference appears to be somewhere in the range of approx. a 5 to 10-fold increase in KDM3A levels 24 hr after detachment. It would be better to show a more representative blot.

3) Loading controls (TUBA) are often not adjusted very carefully making it hard to assess the claimed differences in protein levels. For example this is the case in Figure 2 and Figure 4. This needs to be adjusted.

4) It would be helpful to know how long the cells have been treated with the respective inhibitors in Figure 2 and G.

5) Figure 3: is H3K9me1 changed on the BNIP3 and BNIP3L promoters?

6) ChIP KDM3A and H3K9me1/me2 under KDM3A and KDM3AH1120G/D1122N overexpressing conditions in attached MCF10A cells and perform qRT PCR for BNIP3 and BNIP3L to confirm the results from Figure 1.

7) MF10A detached (96 h) plot in Figure 4—figure supplement 1 shows a smaller second "hump" and thus looks different from the control MF10A detached (96 h) plot in Figure 1—figure supplement 2 and Figure 3—figure supplement 3 suggesting that the cells might have responded abnormally in this experiment. This should be repeated.

*Reviewer #2:*

Green and colleagues in this very interesting study perform a genome-wide RNAi screen to identify new factor that cause anoikis resistance. The study identified a previously unknown transcriptional program regulated by the chromatin modifier protein KDM3A. The manuscript is well written and the data is robust and with appropriate controls. The gene expression data presented in Figure 4—figure supplement 2 also establish the clinical significance of the observed pathway. Overall, this manuscript is a strong candidate for publication in *eLife* as a Brief Report. Addressing the following remaining issues will further strengthen this manuscript.

1) In the Figure 4 the authors show that KDM3A knockdown does not affect the tumor forming ability of the breast cancer cells. However, in Figure 4—figure supplement 8 there seems to difference in tumor growth between NS and KDM3A shRNA. Is this difference significant? On the same line it will be important to include a few lines in the Discussion in regard to the role of anoikis resistane in tumor growth versus metastasis phenotypes. One would imagine that it might also play a role in tumor growth.

2) In Figure 4—figure supplement 2, is it possible to present the data in the context of major breast cancer subtypes (ER+, HER2+, triple negative).

---

## [Author Response]

Reviewer #1:

*[…] In summary, this manuscript not only introduces a novelty aspect by implicating KDM3A in anoikis but also provides the mechanistic and clinical framework to make this an appropriate story for* eLife.

*However, some details are missing that should be addressed:*

1) Is the FAK and EFGR pathway also required to regulate KDM3A on the transcriptional level? Figure 2 only provides evidence that this is the case on the protein level.

To address the reviewer’s comment, we performed qRT-PCR to measure *KDM3A* mRNA levels after treatment of attached MCF10A cells with the same FAK and EGFR inhibitors used in Figure 2, and G. The new results, presented in Figure 2—figure supplement 1 of the revised manuscript, show that treatment with the FAK inhibitor, gefitinib or U0126 resulted in a statistically significant increase in *KDM3A* mRNA, indicating that the FAK and EGFR pathways regulate *KDM3A* at the transcription level.

2) Discrepancy in KDM3A levels between Figure 2 and other westerns throughout the paper. KDM3A levels appear to be virtually absent under attached conditions in Figure 2 whereas generally the difference appears to be somewhere in the range of approx. a 5 to 10-fold increase in KDM3A levels 24 hr after detachment. It would be better to show a more representative blot.

As requested by the reviewer we have replaced the original Figure 2 with another, independent KDM3A immunoblot that is more similar to those in the other figures. In particular, the new immunoblot of Figure 2 shows a low level of KDM3A in attached cells, similar to the other immunoblots in the manuscript. The other immunoblots shown in the manuscript were visualized using a digital imaging system, whereas the original Figure 2 was an older experiment that used a scanned film, which is less sensitive and explains why the weak KDM3A expression in attached cells was not detected.

3) Loading controls (TUBA) are often not adjusted very carefully making it hard to assess the claimed differences in protein levels. For example this is the case in Figure 2 and Figure 4. This needs to be adjusted.

In response to the reviewer’s comment we have replaced the original immunoblots of Figure 2, and G with new immunoblots in which there is not lane-to-lane variability in TUBA levels.

In Figure 4, six different cell lines are analyzed. It is well known that tubulin levels can vary between cell lines (Verdier-Pinard et al. 2003, *Biochemistry* 42:12019**-**12027), which is the basis of the variation of TUBA levels in this figure. In this experiment TUBA levels in the same cell line should be compared and in all cases are relatively equivalent. To avoid possible confusion, in the revised manuscript we have divided the original single immunoblot into six separate immunoblots, each one showing the results of only one cell line

4) It would be helpful to know how long the cells have been treated with the respective inhibitors in Figure 2 and G.

In the original manuscript, this information was provided in the Materials and methods section. In response to the reviewer’s comment, we have also specified the length of inhibitor treatment in the revised legends for Figure 2.

5) Figure 3: is H3K9me1 changed on the BNIP3 and BNIP3L promoters?

To address the reviewer’s comment we performed ChIP analysis monitoring H3K9me1 enrichment on the *BNIP3* and *BNIP3L* promoters in attached versus detached MCF10A cells, and in *KDM3A* knockdown MCF10A cells following detachment. We found that, like H3K9me2, enrichment of H3K9me1 on the *BNIP3* and *BNIP3L* promoters significantly decreased following detachment, which was counteracted by knockdown of *KDM3A*. These new results are presented in Figure 3—figure supplement 2 of the revised manuscript and are consistent with previous studies showing that KDM3A removes both mono-methyl and di-methyl groups from H3K9 (Yamane et al. 2006, *Cell* 125:483-495).

6) ChIP KDM3A and H3K9me1/me2 under KDM3A and KDM3AH1120G/D1122N overexpressing conditions in attached MCF10A cells and perform qRT PCR for BNIP3 and BNIP3L to confirm the results from Figure 1.

We performed all of the experiments requested by the reviewer, which are presented in Figure 3—figure supplement 3 of the revised manuscript. As expected, overexpression of KDM3A resulted in increased occupancy of KDM3A on the *BNIP3* and *BNIP3L* promoters (Figure 3—figure supplement 3). There was also increased occupancy of KDM3A(H1120G/D1122N) on the *BNIP3* and *BNIP3L* promoters, which is not unexpected because the mutations are in the catalytic domain and should not affect the DNA-binding activity (Yamane et al. 2006, *Cell* 125:483-495; Beyer et al. 2008, *J Biol Chem* 283:36542-36552). As expected, overexpression of KDM3A, but not KDM3A(H1120G/D1122N), resulted in decreased levels of H3K9me1 and H3K9me2 on the *BNIP3* and *BNIP3L* promoters. Finally, as expected, overexpression of KDM3A, but not KDM3A(H1120G/D1122N), resulted in increased expression of *BNIP3* and *BNIP3L* as monitored by qRT-PCR (Figure 3—figure supplement 3). In summary, these new results confirm the results of Figure 1, and demonstrate that the inability of the catalytically inactive KDM3A mutant to reduce MCF10A cell viability is due to its inability to remove H3K9me1 and H3K9me2 from the promoters of *BNIP3* and *BNIP3L* and induce their expression.

7) MF10A detached (96 h) plot in Figure 4—figure supplement 1 shows a smaller second "hump" and thus looks different from the control MF10A detached (96 h) plot in Figure 1—figure supplement 2 and Figure 3—figure supplement 3 suggesting that the cells might have responded abnormally in this experiment. This should be repeated.

In these apoptosis experiments, the important measurement is the total percentage of annexin-V positive cells, which is similar (~30%) in the controls (non-silencing (NS) shRNA or uninfected MCF10A cells) in the three experiments mentioned by the reviewer. The “hump” in the histogram noted by the reviewer reflects a subpopulation of apoptotic cells that stain intensely with annexin and has no bearing on the calculation of the total percentage of annexin-V positive cells. It is well known that the annexin-V fluorescent intensity of apoptotic cells can vary substantially (Engeland et al. 1998, *Cytometry* 31:1-9; Prieto et al. 2002, *Cytometry* 48:185-193).

Reviewer #2:

*[…] Overall, this manuscript is a strong candidate for publication in eLife as a Brief Report. Addressing the following remaining issues will further strengthen this manuscript.*

1) In the Figure 4 the authors show that KDM3A knockdown does not affect the tumor forming ability of the breast cancer cells. However, in Figure 4—figure supplement 8 there seems to difference in tumor growth between NS and KDM3A shRNA. Is this difference significant? On the same line it will be important to include a few lines in the Discussion in regard to the role of anoikis resistane in tumor growth versus metastasis phenotypes. One would imagine that it might also play a role in tumor growth.

The differences in primary tumor growth noted by the reviewer are not statistically significant. In response to by the reviewer’s comment, we have added a sentence to the legend for Figure 4—figure supplement 8 explicitly stating that the differences in primary tumor growth between control and *Kdm3a* knockdown cells are not statistically significant. Likewise, we have also added a sentence to the legend for Figure 4 stating that the differences in primary tumor growth are also not statistically significant.

Finally, as requested by the reviewer, we have added a brief statement about the role of anoikis in primary tumor growth to the Introduction of the revised manuscript (page 3).

2) In Figure 4—figure supplement 2, is it possible to present the data in the context of major breast cancer subtypes (ER+, HER2+, triple negative).

The data in Figure 4—figure supplement 2 (Figure 4—figure supplement 4 in the revised manuscript) was obtained from the Oncomine database. For the specific Oncomine studies used in our manuscript information regarding breast cancer subtype is not available. We note that in Figure 4, in which we experimentally analyzed *KDM3A* expression in human breast cancer samples, the results are separated by subtype and show no significant difference between the subtypes. In response to the reviewer’s comment, we have revised the legend for Figure 4 to state that there are no significant differences in *KDM3A* expression between the subtypes.